# A novel nairovirus associated with acute febrile illness in Hokkaido, Japan

Fumihiro Kodama[1,2], Hiroki Yamaguchi[3], Eunsil Park[4], Kango Tatemoto[4], Mariko Sashika[5], Ryo Nakao [6], Yurino Terauchi[7], Keita Mizuma[8], Yasuko Orba [9,10], Hiroaki Kariwa[7], Katsuro Hagiwara[11], Katsunori Okazaki[12], Akiko Goto[3], Rika Komagome[3], Masahiro Miyoshi[3], Takuya Ito[3], Kimiaki Yamano[3], Kentaro Yoshii[13], Chiaki Funaki[9], Mariko Ishizuka[9], Asako Shigeno[14], Yukari Itakura[9], Lesley Bell-Sakyi[15], Shunji Edagawa[1], Atsushi Nagasaka[1], Yoshihiro Sakoda [8], Hirofumi Sawa [9,10,16,17], Ken Maeda[4], Masayuki Saijo [18] & Keita Matsuno [10,14,16✉]

The increasing burden of tick-borne orthonairovirus infections, such as Crimean-Congo hemorrhagic fever, is becoming a global concern for public health. In the present study, we identify a novel orthonairovirus, designated Yezo virus (YEZV), from two patients showing acute febrile illness with thrombocytopenia and leukopenia after tick bite in Hokkaido, Japan, in 2019 and 2020, respectively. YEZV is phylogenetically grouped with Sulina virus detected in *Ixodes ricinus* ticks in Romania. YEZV infection has been confirmed in seven patients from 2014–2020, four of whom were co-infected with *Borrelia* spp. Antibodies to YEZV are found in wild deer and raccoons, and YEZV RNAs have been detected in ticks from Hokkaido. In this work, we demonstrate that YEZV is highly likely to be the causative pathogen of febrile illness, representing the first report of an endemic infection associated with an orthonairovirus potentially transmitted by ticks in Japan.

[1] Sapporo City General Hospital, 060-8604 Sapporo, Japan. [2] Nagaoka Red Cross Hospital, 940-2085 Nagaoka, Japan. [3] Center of Infectious Diseases, Hokkaido Institute of Public Health, 060-0819 Sapporo, Japan. [4] Department of Veterinary Science, National Institute of Infectious Diseases, 162-8640 Shinjuku, Tokyo, Japan. [5] Laboratory of Wildlife Biology and Medicine, Faculty of Veterinary Medicine, Hokkaido University, 060-0818 Sapporo, Japan. [6] Laboratory of Parasitology, Faculty of Veterinary Medicine, Hokkaido University, 060-0818 Sapporo, Japan. [7] Laboratory of Public Health, Faculty of Veterinary Medicine, Hokkaido University, 060-0818 Sapporo, Japan. [8] Laboratory of Microbiology, Faculty of Veterinary Medicine, Hokkaido University, 060-0818 Sapporo, Japan. [9] Division of Molecular Pathobiology, International Institute for Zoonosis Control, Hokkaido University, 001-0020 Sapporo, Japan. [10] International Collaboration Unit, International Institute for Zoonosis Control, Hokkaido University, 001-0020 Sapporo, Japan. [11] School of Veterinary Medicine, Rakuno Gakuen University, 069-8501 Ebetsu, Japan. [12] Laboratory of Microbiology and Immunology, Faculty of Pharmaceutical Sciences, Health Sciences University of Hokkaido, 061-0293 Ishikari-Tobetsu, Japan. [13] National Research Center for the Control and Prevention of Infectious Diseases, Nagasaki University, 852-8521 Nagasaki, Japan. [14] Division of Risk Analysis and Management, International Institute for Zoonosis Control, Hokkaido University, 001-0020 Sapporo, Japan. [15] Department of Infection Biology and Microbiomes, Institute of Infection, Veterinary and Ecological Sciences, University of Liverpool, Liverpool L3 5RF, United Kingdom. [16] One Health Research Center, Hokkaido University, 060-0818 Sapporo, Japan. [17] Global Virus Network, MD 21201 Baltimore, USA. [18] Department of Virology 1, National Institute of Infectious Diseases, 162-8640 Shinjuku, Tokyo, Japan. ✉email: matsuk@czc.hokudai.ac.jp

Tick-borne pathogens are an important cause of morbidity, especially in rural areas where tick–human interactions are increasing. The varied etiologies of acute febrile illness associated with tick bite hamper us in determining the causative agent. Furthermore, an increasing number of newly-identified microbes in ticks highlights the complexity of differentiating the causes of febrile illness occurring after tick bite. Orthonairoviruses (family *Nairoviridae*; genus *Orthonairovirus*) are tick-borne viruses causing sometimes fatal febrile illnesses in humans and other animals. Of 15 species within the genus, four species comprise known human pathogens, including Crimean-Congo hemorrhagic fever virus (CCHFV), Nairobi sheep disease virus, Dugbe virus, and Kasokero virus. Recent identifications of human febrile illness associated with Tamdy virus-like orthonairoviruses, Tacheng tick virus 1 (TcTV1)[1,2] and Songling virus (SGLV)[2] in China, reveal a complex landscape of orthonairovirus infections.

While orthonairoviruses are present in ticks in Japan[3,4], no associated diseases had previously been reported. In the present study, we aimed to identify the causative agent of an acute febrile illness in two cases, characterized by thrombocytopenia, leukopenia, and elevation of liver enzymes and ferritin after presumed tick bite in Hokkaido. To this end, we successfully identified a novel orthonairovirus from the patients. Herein we describe the clinical manifestation of the infection, characteristics of the virus, and epidemiology of the virus in wild animals and ticks. We show endemic infection with a novel orthonairovirus associated with human acute febrile illness by retrospective testing and field study.

## Results

**Patient 1**. Patient 1 was a 41-year-old male with a medical history of hyperuricemia and hyperlipidemia, who lived in Sapporo, Hokkaido. In mid-May 2019, he visited a forest area near Sapporo for approximately 4 h. The next day, he noticed and removed an arthropod attached to his right abdomen. Four days after visiting the forest, he had a fever over 39 °C, followed by gait disturbance and leg pain. After the fever continued for 4 days, he was admitted to our hospital with a temperature of 38.9 °C (Table 1). On admission, a review of systems was negative except for a fever, appetite loss, and bilateral lower extremity pain. Physical examination was unremarkable except for a small papule on the abdomen, which was suspected to be the site of a tick bite. Laboratory tests showed leukopenia and lymphocytopenia with a white-blood cell (WBC) count of 1,600/μl and a lymphocyte ratio of 24.0% on differential count (Fig. 1a), thrombocytopenia with a platelet count of 87,000/μl (Fig. 1b), and significantly increasing levels of aspartate aminotransferase (AST, 3703 units/l), alanine aminotransferase (ALT, 1,783 unit/l), creatine kinase (CK; 5,847 unit/l), lactose dehydrogenase (LDH, 4,069 unit/l) (Fig. 1c), and ferritin (55,200 ng/ml) (Supplementary Table 1). Computed

tomography scans of the chest, abdomen, and pelvis with contrast showed findings consistent with fatty liver disease without organomegaly.

During hospitalization, thrombocytopenia progressed until day 6 (6 days after the onset of fever, the second day in hospital) when the lowest platelet count reached 75,000/μl. This was followed by thrombocytosis and a peak platelet count of 638,000/μl. Atypical lymphocytosis was observed throughout the hospitalization period with the peak level of atypical lymphocytes reaching 76% in a WBC count of 8,200/μl. Significant elevation of AST, ALT, LDH, and CK levels lasted until day 10. Activated partial thromboplastin time (APTT) was extended on days 5–9 (up to 52 s on day 8), and elevations of fibrinogen degradation products (FDP; maximum of 14.1 μg/ml) and D-dimer (maximum of 6.6 μg/ml) were recorded on day 6 (Fig. 1d).

On the basis of the history of a suspected tick bite, the patient was treated empirically with 8 days of ceftriaxone for suspected Lyme disease or *Borrelia miyamotoi* infection, 14 days of doxycycline for suspected rickettsioses, and 6 days of gentamicin for suspected tularemia. His fever and other symptoms gradually resolved, and the patient was discharged without complications after day 19 (15 days of hospitalization).

The patient's sera, collected on days 5 (the day of hospitalization) and 19, were negative on both days for IgM and IgG antibodies for Lyme disease (the *Borrelia* spp.) and relapsing fever (i.e., *B. miyamotoi*), neutralizing antibodies for tick-borne encephalitis virus (TBEV) and Japanese encephalitis virus, and specific antibodies for *Francisella tularensis* and *Rickettsia* spp. Polymerase chain reaction (PCR) assays were performed on whole blood taken on day 5; these assays also were negative for *Borrelia* spp. and severe fever with thrombocytopenia syndrome virus (SFTSV).

A skin biopsy around the papule of the suspected tick bite was performed on the second day of hospitalization, and a fragment of the *gltA* gene of *R. helvetica* was detected in this specimen.

**Patient 2**. Patient 2 was a 59-year-old previously healthy male with no remarkable medical history living in Sapporo, Hokkaido. In mid-July 2020, he hiked on a mountain near Sapporo. During the hike, he received a bite on his lower leg from an unidentified arthropod that remained attached for at least 30 min. He remained in his usual state of health until 9 days after the hike when he lost his appetite and then, developed a fever of 37.4 °C on 17 days after the hike. Following two visits to different hospitals on days 3 and 4 after the onset of fever (Table 1), where he was found to have a fever (38.5 °C on day 3) with leukopenia (WBC count 1,170/μl and lymphocytes at 38.0% on differential count, Fig. 1a) and thrombocytopenia (platelet count 65,000/μl, Fig. 1b), he visited our hospital on day 5 post-onset of fever. Our laboratory tests on day 5 confirmed progressing thrombocytopenia (52,000/μl), continuing leukopenia (WBC count 1700/μl and lymphocytes at 38.0% on

**Table 1 Summary of two patients.**

| Patient (ID) | Medical history | Symptoms at presentation | Laboratory findings | Tick-borne pathogen detection (result) |
|---|---|---|---|---|
| Patient 1 (HH001-2019) | Hyperuricemia Hyperlipidemia | Fever (38.9 °C) Appetite loss Bilateral lower extremity pain | Leukopenia Lymphocytopenia Thrombocytopenia Increasing AST, ALT, LDH, CK, ferritin Coagulation disorder | *Borrelia* spp. (−) *Francisella tularensis* (−) *Rickettsia* spp. (−) TBEV (−) SFTSV (−) |
| Patient 2 (HH003-2020) | — | Fever (38.5 °C) Appetite loss Pruritic urticarial rash on extremities | Leukopenia Thrombocytopenia Increasing AST, ALT, LDH, CK, ferritin Coagulation disorder | *Borrelia miyamotoi* (+) *Borrelia burgdorferi* (+) TBEV (−) SFTSV (−) |

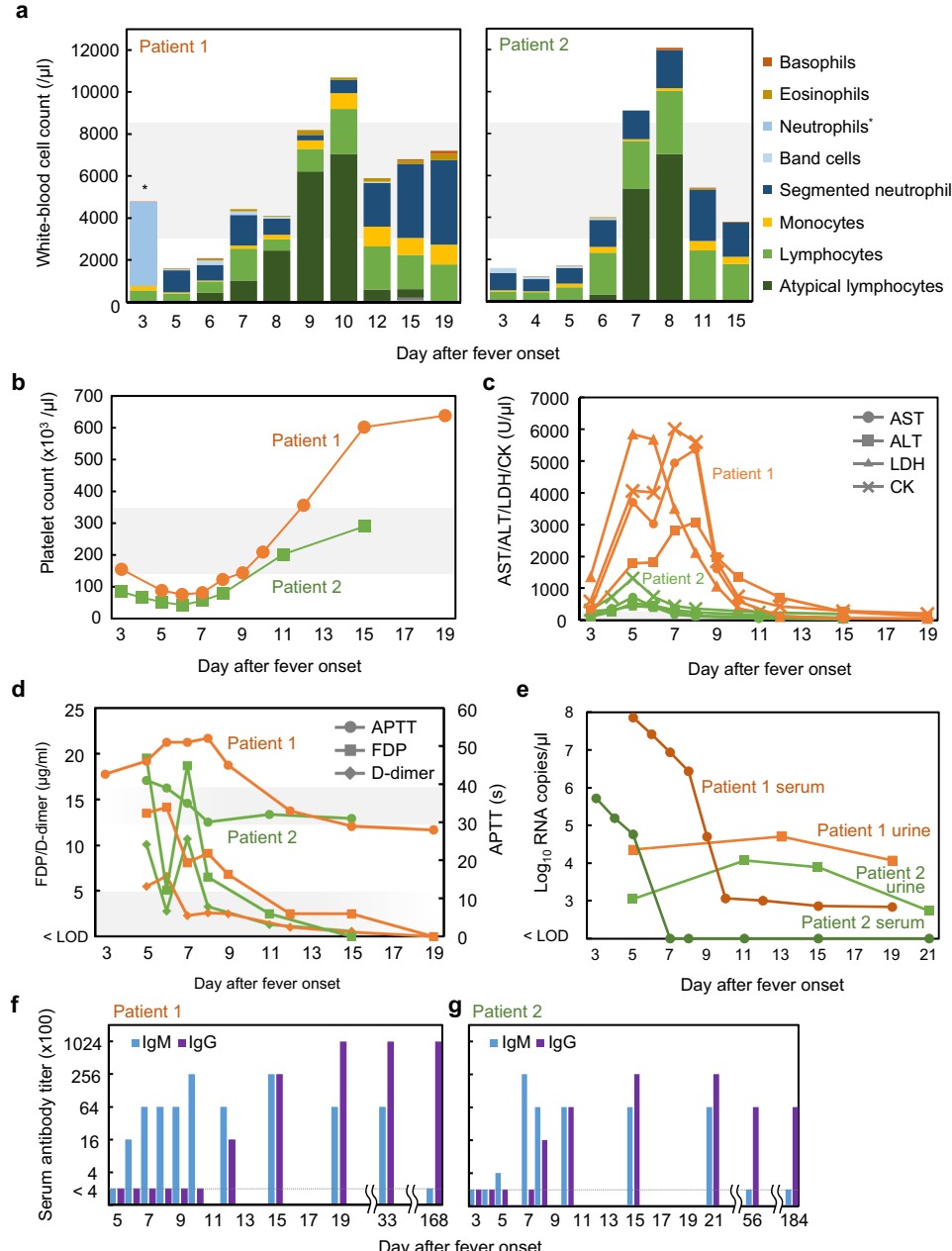

**Fig. 1 Laboratory test values of two patients infected with Yezo virus (YEZV). a** White-blood cell count shown as the total of the differential counts. Cell types not indicated in the key were undetectable. *Neutrophils were further differentiated into band cells and segmented neutrophils except for the sample from patient 1 on day 3. Gray shading behind the graphs indicates the approximate normal ranges of values. **b** Platelet counts for patient 1 (orange) and patient 2 (light green). **c** Aspartate aminotransferase (AST, circle), alanine aminotransferase (ALT, square), lactose dehydrogenase (LDH, triangle), and creatinine kinase (CK, cross) values for patients 1 (orange) and 2 (green). **d** Activated partial thromboplastin time (APTT, circle), fibrinogen degradation products (FDP, square), and D-dimer (rhombus) plotted for patients 1 (orange) and 2 (green). **e** Viral loads determined for serum (dark color with circle) and urine (regular color with square) for patients 1 (orange) and 2 (green) using RT-qPCR. Limit of detection (LOD) was approximately 100 copies/μl. **f, g** Serum antibodies reacting with YEZV N protein were detected using enzyme-linked immunosorbent assay (ELISA) for patients 1 (**f**) and 2 (**g**). The reciprocal of the highest serum dilution in which the difference in OD value between YEZV N and the negative control was >0.3, was determined as the serum antibody titer. Anti-human IgM (light blue) or IgG (dark blue) were used to detect specific antibodies binding to the antigen.

differential count), and elevated AST, ALT, LDH, ferritin, and CK levels (Fig. 1c and Supplementary Table 1). On day 6, pruritic urticarial rash developed on his extremities (Supplementary Fig. 1), but the fever resolved.

During his visits to the hospitals, thrombocytopenia progressed to a minimum of 42,000/μl on day 6. Lymphocytopenia began to resolve on day 5 and atypical lymphocytosis appeared with the atypical lymphocyte population increasing to 58% on day 8.

Elevation of AST, ALT, LDH, and CK levels continued to resolve after day 8. An extended APTT (up to 41 s on day 5) and elevations of FDP (maximum of 19.5 μg/ml on day 5) and D-dimer (maximum of 10.7 μg/ml on day 7) were observed by day 7 (Fig. 1d).

The patient was administered doxycycline after the second hospital visit on day 4. The patient's blood was collected on day 5 and tested positive for *B. miyamotoi* and negative for

*B. burgdorferi* by PCR, while serum collected on day 7 was positive for *B. miyamotoi* IgG antibody and Lyme disease *Borrelia* IgM and IgG antibodies. RNAs of TBEV and SFTSV were not detected in serum collected on day 5.

**Detection of a novel orthonairovirus in patients**. A serum sample from patient 1 on day 4 was initially used for virus isolation using Vero E6 cells. After 14 days without any cytopathic effect (CPE), RNA fragments of a possible orthonairovirus, Yezo virus (YEZV) tentatively named after the historical name of Hokkaido, were detected from the supernatant using high-throughput sequencing. Then, primers for reverse-transcription PCR (RT-PCR) and quantitative RT-PCR (RT-qPCR) were designed to detect the YEZV RNA fragments.

Serum and urine samples from the two patients were subjected to RT-qPCR to reveal the viral load during febrile illness (Fig. 1e). The highest viral load observed was $7.3 \times 10^7$ copies/μl in the serum of patient 1 at day 5. Serum samples collected from patient 2 during the acute phase of infection were also positive for YEZV RNA, with the highest value being $5.2 \times 10^5$ copies/μl. The serum viral load dropped steeply at days 7–8 in patient 1 and at days 5–6 in patient 2. YEZV RNA was detectable in the serum of patient 1 until day 19. Urine viral loads were lower than serum viral loads during the acute phase of infection but were maintained even after the steep drop in the serum viral load in patients 1 and 2. Serum and/or urine were negative for YEZV by RT-qPCR after day 33 for patient 1 and day 56 for patient 2.

Antibody levels in patient sera were measured using enzyme-linked immunosorbent assay (ELISA), with YEZV nucleocapsid (N) protein-expressing cells as the antigen. IgM antibodies were detected at day 6 in patient 1 (Fig. 1f) and day 5 in patient 2 (Fig. 1g) followed by a peak IgM antibody titer of 25,600 and disappearance approximately 6 months later (day 168 for patient 1 and day 184 for patient 2). IgG antibody titer increased on day 11 in patient 1 and day 7 in patient 2 and was still high at the end of the study period.

**Retrospective identification of patients infected with YEZV.** We retrospectively screened serum samples of 248 patients suspected as having a tick-borne disease since 2014 using RT-PCR and RT-qPCR assays. Five positive samples were identified and were tested by ELISA, along with two available convalescent serum samples in conjunction with the positive samples. In total, seven patients, including patients 1 and 2, were confirmed to be infected with YEZV in the acute phase of febrile illness with thrombocytopenia after tick bite, four of whom were found to have seroconverted from undetectable levels of IgG at their acute phase to more than 6,400 ELISA titers at their convalescent phase (Table 2). Four out of the seven patients tested positive for *Borrelia* spp. infections.

**Isolation and genetic identification of YEZV.** After the identification of YEZV RNA fragments in Vero E6 cells, no further growth of the virus was confirmed during consecutive passages of the samples. Then, sequential passage of the virus from patient 2 through AG129 mice and Vero E6 cells was performed. While two AG129 mice intraperitonially inoculated with a plasma sample did not show clinical signs until 14 days, seroconversion with an ELISA IgG titer of 409,600 was identified. Swelling of the spleen was found in another two mice 5 days after inoculation when they were sacrificed for serum collection. The serum samples were passaged onto Vero E6 cells and infectious YEZV was recovered without causing any CPE in Vero E6 cells. Spherical and oval virions of approximately 100–200 nm in diameter at the

**Table 2 Summary of clinical findings and pathogen detection in patients infected with YEZV.**

| Patient ID | Sex | Age | Date | Tick bite | Fever | Joint/muscle pain | Neurological sign/paralysis/numbness | Increasing liver enzymes | Thrombocytopenia | Leukocytopenia | YEZV infection RNA | YEZV infection IgG | Borrelia infection B. miyamotoi | Borrelia infection B. burgdorferi sensu lato |
|---|---|---|---|---|---|---|---|---|---|---|---|---|---|---|
| HH004-2014 | F | 60 s | 2014-May | + | + | : | : | + | + | + | + | NA | ND | ND |
| HH007-2016 | M | 20 s | 2016-July | + | + | : | : | + | + | + | + | 6,400 (24 d) | + | ND |
| HH008-2017 | M | 30 s | 2017-June | + | 39 °C | + | + | : | + | : | + | NA | ND | ND |
| HH009-2017 | F | 70 s | 2017-June | + | 38.5 °C | + | : | : | + | + | + | NA | ND | + |
| HH001-2019 (Patient 1) | M | 41 | 2019-May | + | 39 °C | + | + | + | + | + | + | 102,400[a] | ND | ND |
| HH011-2020 | M | 80 s | 2020-May | + | 38 °C | : | : | + | + | + | + | 6,400 (36 d) | + | + |
| HH003-2020 (Patient 2) | M | 59 | 2020-July | + | 38.5 °C | : | + | + | + | + | + | 25,600[a] | + | + |
| Total | | | | 7/7 | 7/7 | 3/7 | 3/7 | 5/7 | 7/7 | 6/7 | 7/7 | 4/7 | 4/7 | 4/7 |
| Percentage | | | | 100.0% | 100.0% | 42.8% | 42.8% | 71.4% | 100.0% | 85.7% | 100.0% | 57.1% | 57.1% | 57.1% |

*NA* not available, *ND* not detected
[a]Shown in Fig. 1f

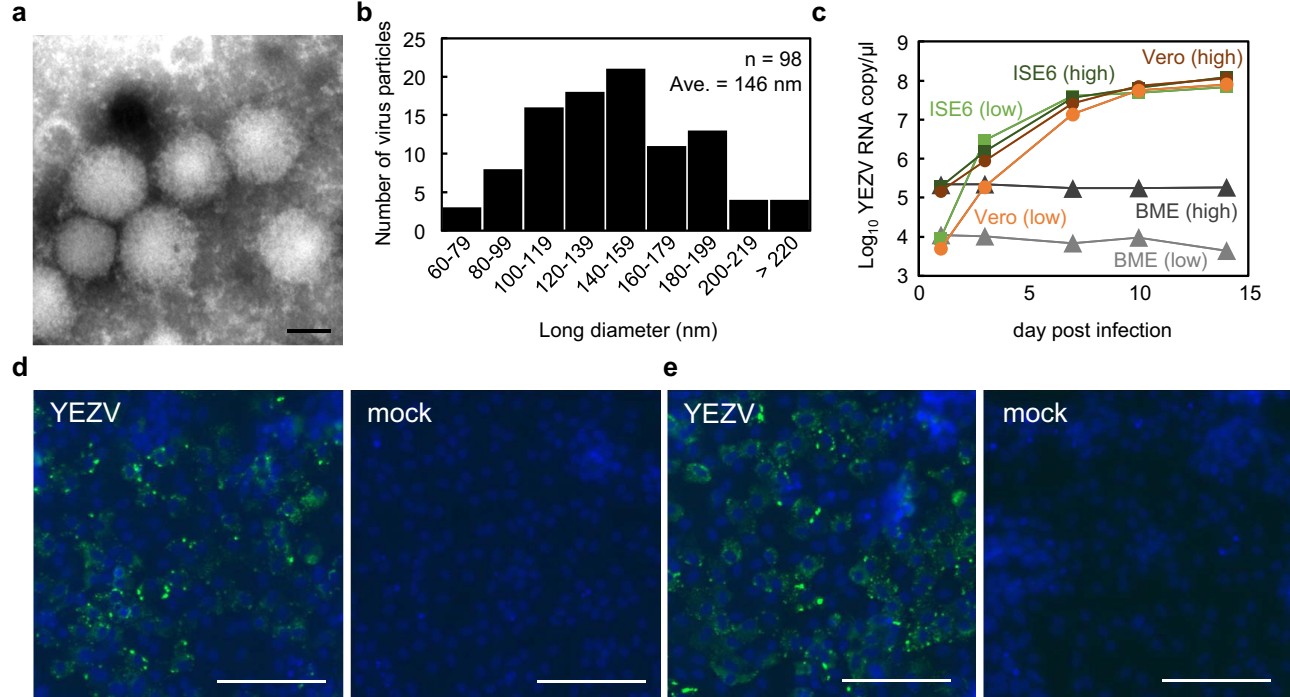

**Fig. 2 Isolation of a novel orthonairovirus, YEZV. a** Transmission electron microscopy of YEZV particles negatively stained with 2% phosphotungstic acid (pH 7.0). Scale bar = 100 nm. Similar particle images were obtained in two independent experiments. **b** Particle size distribution shown by the long diameters of spherical and oval virions in the microscopy images. Diameters of 98 particles were manually measured, and the number of particles in each 20-nm range are shown. **c** Growth of YEZV in different cell lines. YEZV was inoculated at a multiplicity of infection of 0.1 (high, dark color) or of 0.001 (low, regular color) into mammalian-derived Vero E6 cells (orange), tick ISE6 (green) and BME/CTVM23 (BME, gray) cells, and incubated for 14 days. Viral RNA copies at each time point were measured using RT-qPCR. **d**, **e** Detection of YEZV antigens in Vero E6 cells. Vero E6 cells were infected with YEZV and fixed after seven days. Convalescent serum collected from patient 1 on day 168 (**d**) and from patient 2 on day 184 (**e**) after the onset of fever, were used in an immunofluorescent assay. Antigens reacting with human IgG are shown in green. Cell nuclei are shown in blue, scale bar = 100 μm. Similar results were obtained in two independent experiments.

widest point were observed in the supernatant of the infected Vero E6 cells by transmission electron microscopy (Fig. 2a, b, and Supplementary Fig. 2). *Ixodes scapularis* embryo-derived ISE6 cells[5] supported the replication of YEZV to titers similar to those in Vero E6 cells, while no viral RNA replication was detected in *Rhipicephalus microplus* embryo-derived BME/CTVM23 cells[6] over 14 days (Fig. 2c). Immunofluorescent staining with convalescent sera from patients 1 and 2 revealed that viral antigens were detected in the cytoplasm of Vero E6 cells infected with YEZV (Fig. 2d, e).

Sequence analysis of RNA extracted from culture supernatant identified entire YEZV RNA sequences constituting three genome segments: large (L), medium (M), and small (S) (Fig. 3a). Each genome segment encoded a single open reading frame whose deduced amino acid sequence was similar to each corresponding orthonairovirus protein, i.e., RNA-dependent RNA polymerase (L protein), glycoprotein precursor (GPC), and N protein. Sequences obtained from patients are shown in Supplementary Table 2. The sequences of the YEZV strains were nearly identical to each other (>97% at the nucleotide level and >99% at the protein level). All three RNA segments of YEZV were phylogenetically grouped with Sulina virus, which was discovered in *I. ricinus* ticks collected in Romania[7] (Fig. 3b–d). Sequence identity between YEZV and Sulina virus was 70.1% (L segment), 58.4% (M segment), and 54.1% (S segment), and 82.2% (L protein), 56.7% (GPC), and 60.2% (N protein). Based on the high sequence identity of the N protein sequence between YEZV and Sulina virus, YEZV will be assigned to the genogroup Sulina[8].

**Detection of antibodies to YEZV in wild animals and YEZV RNA in ticks.** To investigate the possible natural reservoir of YEZV in Hokkaido, serological screening was performed with serum samples collected from wild animals in Hokkaido between 2010 and 2020 (Table 3). Among Hokkaido shika deer (*Cervus nippon yesoensis*) and raccoons (*Procyon lotor*) in Hokkaido, 6/785 (0.8%) and 3/182 (1.6%) animals were found to be positive for YEZV antibodies, respectively, while no Hokkaido raccoon dogs (*Nyctereutes procyonoides albus*) or rodents (*Myodes rufocanus bedfordiae* and *Apodemus speciosus*) tested positive. The three major tick species (i.e. *Haemaphysalis megaspinosa*, *Ixodes ovatus*, and *Ixodes persulcatus*), collected from vegetation in Hokkaido in 2016–2020, were screened for YEZV RNA using RT-qPCR (Table 4). The highest detection rate of YEZV among these three species was 3.8% in *H. megaspinosa*.

**Discussion**

Although the complete fulfillment of Koch's postulates has not yet been achieved to prove the identity of the causative agent of the tick-borne infectious disease in Hokkaido, Japan, our findings are compatible with the identification of an emerging pathogenic virus, YEZV. Before the discovery of YEZV, Lyme disease, *B. miyamotoi* infection, and TBE were the known tick-borne infections affecting humans in Hokkaido. While co-infection with *Borrelia* spp. was common in our patients, other tick-borne infections such as rickettsioses, TBE, and SFTS were excluded. We demonstrated that at least seven patients were infected with YEZV since 2014, and that wild animals and ticks may be

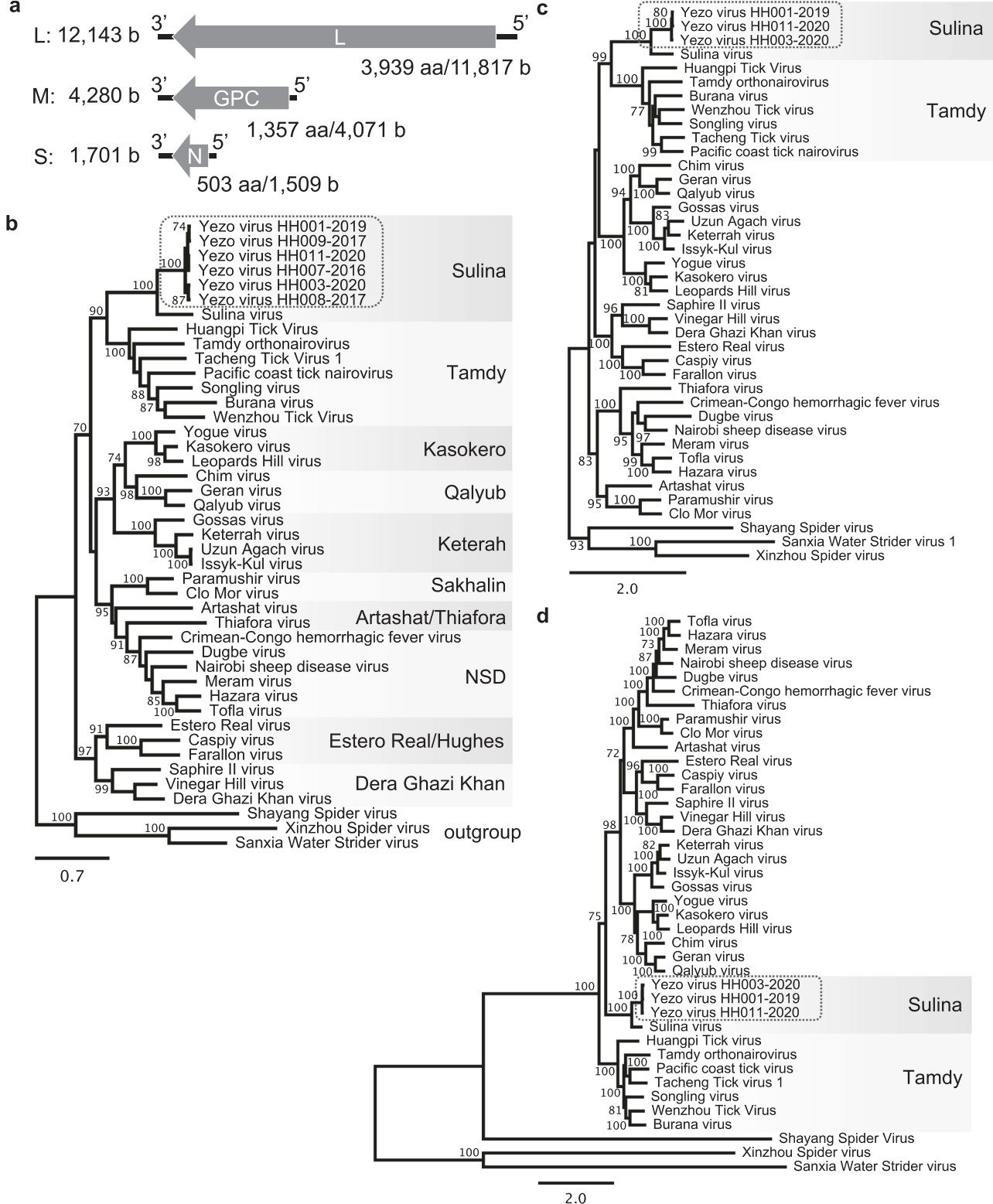

**Fig. 3 Genetic characterization of YEZV. a** Schematic diagram of YEZV genomic RNA segments. The YEZV genome comprises three RNA segments, L, M, and S. Each segment encodes a single open reading frame, as indicated by an arrow: L protein by L segment, GPC by M segment, and N protein by S segment. Phylogenetic relationships of (**b**) the N protein, (**c**) the GPC, and (**d**) the L protein coding sequences of orthonairoviruses including YEZVs, as shown surrounded by a broken line. Nucleotide sequences of orthonairovirus coding sequences available in public databases were aligned together with those of YEZV strains. The trees were constructed using the maximum-likelihood method with 1000 bootstraps. Branch support (%) higher than 70% is indicated beside each branch. Orthonairovirus genogroups are indicated to the right of the trees.

**Table 3 Detection of YEZV antibodies in serum samples from wild animals.**

|  | Duration | Total | Positive | Positive rate |
|---|---|---|---|---|
| Hokkaido shika deer | 2010–2019 | 785 | 6 | 0.8% |
| Raccoon | 2017–2020 | 182 | 3 | 1.6% |
| Hokkaido raccoon dog | 2017–2020 | 125 | 0 | 0.0% |
| Rodent | 2019 | 41 | 0 | 0.0% |

**Table 4 Detection of YEZV RNA in adult ticks.**

|  | Duration | Total | Positive | Positive rate |
|---|---|---|---|---|
| Total |  | 477 | 10 | 2.1% |
| *Haemaphysalis megaspinosa* | 2016–2020 | 108 | 4 | 3.7% |
| Female |  | 55 | 1 | 1.8% |
| Male |  | 53 | 3 | 5.7% |
| *Ixodes ovatus* | 2016–2020 | 213 | 4 | 1.9% |
| Female |  | 131 | 1 | 0.8% |
| Male |  | 82 | 3 | 3.7% |
| *Ixodes persulcatus* | 2016–2020 | 156 | 2 | 1.3% |
| Female |  | 87 | 2 | 2.3% |
| Male |  | 69 | 0 | 0.0% |

potential reservoirs for the virus, suggesting that YEZV infection is endemic in this area. Furthermore, the possible role of *Ixodes* ticks in the transmission of YEZV in addition to other tick-borne pathogens may highlight global burden of the emerging orthonairovirus.

The symptoms of YEZV infection in two patients were relatively nonspecific, such as fever and malaise. Patient 1 had difficulty walking due to bilateral lower extremity pain, probably secondary to ongoing YEZV-associated myositis. Although patient 2 had transient self-reported gait disturbance and left-hand weakness, no abnormal neurological findings consistent with his symptoms were found. It is unclear whether YEZV infection is associated with peripheral neuropathy or neuritis. Thrombocytopenia, leukopenia, and elevated liver enzyme levels were common in laboratory testing in patients diagnosed with YEZV infections. In addition, highly elevated serum ferritin and CK levels, and significant atypical lymphocytosis, were noted in two patients, and these findings may be specific to YEZV infection. The clinical presentation in these two patients highlighted the various manifestations of YEZV infections: patient 1 had a significantly high serum viral load, as well as elevated liver enzymes, ferritin, and CK levels, while patient 2 had relatively mild to moderate disease with lower viral loads. The neurological symptoms and pruritic urticarial rash observed with patient 2 could be due to co-infection with *Borrelia* spp. The clinical features in the seven patients were partly comparable with those of other tick-borne diseases endemic in Japan and neighboring countries, such as Lyme disease[9,10], *B. miyamotoi* infection[11], and SFTS[12,13]. Nevertheless, gastrointestinal tract symptoms, which are frequently observed in SFTS patients[12,14], were not recorded in the two patients in the present study. While our study was limited to Hokkaido, where the endemicity of SFTS has not been proven[15,16], the overlapping distribution of YEZV and SFTSV in southern Japan is still unknown. Thus, the differential diagnosis between these emerging tick-borne bunyavirus infections should be established with a higher number of patients infected with YEZV.

Because patients with suspected tick bites are routinely administered antibiotics, as were patients 1 and 2, the association between a clinical outcome and a possible bacterial pathogen may be unclear. Our study showed that at least three patients lacked laboratory evidence of a concomitant infection with tick-borne *Borrelia* spp., suggesting that YEZV infection might be the only culprit responsible for acute febrile illness with leukopenia and thrombocytopenia; the remaining four patients were coinfected with YEZV and *Borrelia* spp. Determining the main causative pathogen of clinical features during coinfection of tick-borne pathogens is required for diagnosis because ticks are capable of carrying and transmitting multiple microorganisms simultaneously[17–20]. Establishment of animal models to recapitulate clinical outcomes of YEZV infection is needed to verify the true impact of coinfection of YEZV and *Borrelia* spp.

While the seroprevalence of YEZV in wild animals was lower than that of another tick-borne bunyavirus in Hokkaido[21], our field studies on wild animals and ticks suggested endemic circulation of YEZV or YEZV-like viruses in the local area. Because of potential cross-reactivity of orthonairovirus N antigens[22], a neutralization test using YEZV should be performed in the future to exclude the possibility of infections of a novel virus antigenically cross-reactive to YEZV. The detection of *R. helvetica* DNA in patient 1 and the coinfections of *Borrelia* spp. in patients suggest the association of *Ixodes* ticks, especially *I. persulcatous*, with YEZV infection[23]. Efficient replication of YEZV in *I. scapularis* (blacklegged tick)-derived cells also implied the importance of *Ixodes* ticks as vectors of YEZV. However, no cell lines are available from *Haemaphysalis* spp. ticks to confirm the susceptibility of this tick genus to the virus[24]. Because ticks belonging to the *I. ricinus* species complex, including *I. ricinus* ticks, which harbor Sulina virus, *I. persulcatus*, and *I. scapularis*, are widely distributed in the northern hemisphere, possible endemic occurrences of YEZV and similar orthonairovirus infections should be examined globally, together with other human pathogens transmitted by *Ixodes* ticks, such as species of *Borrelia*, *Babesia*, *Rickettsia*, and TBEV[18].

YEZV is the first cultured virus in the genogroup Sulina. Basic in vitro characterizations of the YEZV isolate, i.e. electron microscopy, growth kinetics, and detection of virus antigens, have been reported in the present study. Future investigations of YEZV both in vitro and in vivo are required to understand the molecular basis of the pathogenesis of YEZV. Furthermore, because the epidemiology and pathogenicity of Tamdy and Sulina group viruses are not yet fully elucidated, these emerging viruses in Asian countries could represent a larger burden on public health than currently recognized. Comprehensive understanding of orthonairoviruses is highly desired to develop prophylaxes and therapeutics.

## Methods

**Cells**. Vero E6 (JCRB9007), Huh-7 (JCRB0403), and THP-1 (JCRB0112) cells were obtained from the Japanese Collection of Research Bioresources Cell Bank. HEK293T cells (CRL-3216) were obtained from the American Type Culture Collection. Cells were maintained using Dulbecco's Minimum Essential Medium (DMEM, Thermo Fisher Scientific) supplemented with 10% fetal bovine serum (FBS), 100 unit/ml penicillin, and 100 μg/ml streptomycin at 37 °C with 5% $CO_2$. The ISE6 and BME/CTVM23 tick cell lines were provided by the CEH Institute of Virology and Environmental Microbiology (Oxford, UK) and the Tick Cell Biobank at the University of Liverpool, respectively. Use of ISE6 cells was authorized by the University of Minnesota, the original provider of the cell line. The ISE6 cells were cultured in L-15B medium supplemented with 10% FBS, 5% tryptose phosphate broth (TPB, Merk) and 0.1% bovine lipoprotein concentrate (MP Biomedicals) at 32 °C[25]. The BME/CTVM23 cells were maintained in L-15 (Leibovitz) medium supplemented with 20% FBS, 10% TPB and 2 mM L-glutamine at 32 °C[6].

**Isolation of virus from patient's blood**. The isolation of virus was initially attempted using serum samples from patients 1 and 2 as follows. Serum and urine samples collected on day 5 and diluted 1:100 in serum-free DMEM were inoculated onto Vero E6, Huh-7, and differentiated THP-1 cells precultured with 50 ng/ml of

phorbol 12-myristate 13-acetate in the growth medium for 2 days prior to the inoculation. Then, after 1 h of the inoculation, the medium was changed to DMEM supplemented with 2% FBS. Inoculated cells were monitored for 2 weeks, then the supernatant was collected and passaged onto freshly prepared cells. However, this method failed to obtain infectious virions after three blind passages.

The second isolation attempt was performed using a plasma sample collected from patient 2 on day 4. The plasma, diluted 1:100 in PBS, was inoculated intraperitoneally into two 9-week-old female AG129 mice (double knockout mice of interferon α/β and γ receptors, obtained from Marshall BioResources, in-house breeding). Mouse sera were collected on day 5 after inoculation, diluted 1:100 in serum-free DMEM, and inoculated onto Vero E6 cells. Three weeks after inoculation, the supernatant was harvested and passaged twice in freshly prepared Vero E6 cells. Cells in the third passage were used for immunostaining and the supernatant collected from the same passage was used for ultracentrifugation and electron microscopy, as described below. The growth of the virus was monitored in mouse serum and supernatant using RT-PCR or RT-qPCR.

### Growth kinetics of YEZV in cells.
YEZV was inoculated into Vero E6, ISE6, and BME/CTVM23 cells at multiplicities of infection (MOI) of approximately 0.1 (high) and 0.001 (low) determined by focus forming units per cell as described below, diluted in serum-free DMEM or complete growth medium of tick cells, as appropriate. After a 1-h incubation on Vero E6 cells, cells were washed once with serum-free DMEM and incubated in DMEM 2% FBS. For ISE6 and BME/CTVM23 cells, the inoculum was removed after a 1-h incubation and was replaced with complete growth medium. YEZV-infected cells were monitored for 14 days.

### Focus forming assay of YEZV.
Vero E6 cells were inoculated with YEZV serially diluted 10-fold in serum-free DMEM. After incubation at 37 °C for 1 h, the inoculum was removed and cells were washed once with serum-free DMEM. Then, Eagle's Minimum Essential Medium supplemented with 2% FBS, 100 units/ml penicillin, 100 μg/ml streptomycin, and 1% methyl cellulose 4000 (Fujifilm Wako) was overlayed on the cell monolayer. Cells were fixed with 10% formalin and washed thoroughly with PBS. Following permeabilization using 0.1% Triton X-100 in PBS for 5 min, cells were reacted with patient convalescent serum diluted 1:1,000 in PBS, followed by washing with PBS three times. Virus antigens were detected using rabbit anti-human IgG (H + L) conjugated with horseradish peroxidase (Abcam) and SIGMAFAST 3,3′-diaminobenzidine tablets (Merck). Visible foci were counted manually for determining focus forming units in the original YEZV stock.

### Ultracentrifugation and transmission electron microscopy.
Approximately 50 ml of the supernatant of Vero E6 cells infected with YEZV at passage three was harvested 7 days post-inoculation and centrifuged at $3500 \times g$ for 10 min at 4 °C to remove debris. Then, the supernatant was ultracentrifuged at $96,589 \times g$ for 2 h at 4 °C using an SW32Ti rotor in an Optima L-90K centrifuge (Beckman Coulter). The pellet was resuspended in 100 μl of PBS and incubated overnight at 4 °C. Concentrated virions in the resuspended pellet were adsorbed onto an ion-sputtered grid with carbon film (Nisshin EM) for 1 min and stained with 2% phosphotungstic acid solution (pH 7.0) for 15 s. Images of the virions were obtained using a transmission electron microscope (H-7650, Hitachi High-Tech). The long diameters of the virions were set manually and were measured using Fiji software[26]. Excel for Mac (Microsoft) was used for displaying the histogram.

### Immunostaining of YEZV-infected cells.
Vero E6 cells infected with YEZV at passage three were fixed with 10% formalin for 30 min at 7 days post-inoculation and were then rinsed twice with PBS. After permeabilization using 0.1% Triton X-100 in PBS for 5 min, cells were reacted with patient convalescent serum diluted 1:1,000 in PBS, followed by washing with PBS three times. Antigens were visualized using goat anti-human IgG (H + L) conjugated with Alexa Fluor 488 (Thermo Fisher Scientific) diluted 1:1,000 in PBS, followed by counter staining using -Cellstain- DAPI solution (DOJINDO) and washing with PBS three times. Images were obtained using a ZOE fluorescent cell imager (Bio-Rad).

### RNA extraction, RT-PCR, and RT-qPCR.
RNA was extracted from patient samples and cell supernatants using TRIzol LS (Thermo Fisher Scientific) or NucleoSpin RNA (Takara) according to the manufacturers' protocols. RNA was extracted from individual host-questing adult ticks using a blackPREP tick DNA/RNA kit (Analytik-Jena) and stored at −80 °C until use. Detection of YEZV S segment RNA was performed using the PrimeScript One Step RT-PCR kit ver. 2 (Takara), forward primer 5′-TGCTCCAATCCCAGAATGTGCCTGG-3′, and reverse primer 5′-CCTGTGCCTTCTCTTGCTCCTCATGTC-3′, under the following thermal conditions: 50 °C for 30 min and 94 °C for 2 min followed by 35 cycles of 94 °C for 30 s, 60 °C for 30 s, and 72 °C for 30 s, plus an additional DNA extension step of 72 °C for 5 min. The amounts of YEZV L segment RNAs were quantified using the One Step PrimeScript III RT-qPCR mix (Takara) and PrimeTime qPCR assays (primer 1: 5′-TCAACCTGCTTCCAACCTATC-3′, primer 2: 5′-CACCCGTACCACAAGAGAATTA-3′, and probe: 5′/5HEX/ACCAAGGAA/ ZEN/GCACACAGATGGGT/3IABkFQ/) (Integrated DNA Technologies) on a StepOnePlus Real-Time PCR system (Thermo Fisher Scientific) with the following thermal conditions; 52 °C for 5 min, 95 °C for 10 s, and 40 cycles of 95 °C for 5 s

and 60 °C for 30 s followed by measurement. A standard curve was generated using a series of diluted plasmids carrying a fragment of YEZV L segment cDNA, and the RNA copy number in each sample was calculated.

### High-throughput sequencing and determination of YEZV complete genome sequences.
Using RNA samples extracted from cell supernatant, sequencing libraries were prepared using the KAPA RNA hyper prep kit (for Illumina) and the KAPA dual-indexed adapter kit (Roche). The prepared MiSeq libraries were sequenced on an MiSeq sequencer using the MiSeq reagent kit v3 (Illumina) with $2 \times 300$-bp paired-end read length. Sequencing reads were de novo assembled using the CLC genomics workbench ver. 20 (Qiagen). Nucleotide sequences of the three assembled RNA segments of YEZV were confirmed by RT-PCR and Sanger sequencing using primers indicated in Supplementary Table 3. The terminal nucleotide of YEZV isolate HH003-2020 was determined using the method described for paramyxovirus genomes[27] with YEZV gene-specific primers (Supplementary Table 4). Briefly, DT88 linker was ligated to the 5′ end of the viral RNA using T4 RNA ligase (NEB) followed by reverse transcription using DT89 with Super Script III (Thermo Fisher Scientific) and PCR using DT89 and a gene-specific primer with KOD One PCR master mix (TOYOBO) for 5′-RACE. For 3′-RACE, DT88 linker was ligated to the 5′ end of the viral cDNA using T4 RNA ligase followed by PCR using DT89 and a gene-specific primer. Using primers designed for YEZV HH003-2020 (Supplementary Table 3), complete genome sequences for two strains and complete S segment sequences for three strains were finally obtained. All sequences analyzed in the present study were deposited in the DNA Data Bank of Japan (DDBJ) under the accession numbers indicated in Supplementary Table 2.

### Phylogenetic analysis.
Coding sequences of the YEZV genome and sequences available in GenBank (Supplementary Table 2) were extracted and assembled using translation align in Geneious Prime 2021.1.1 (Biomatters) with Clustal Omega[28]. Multiple sequence alignments were then manually curated and used to construct phylogenetic trees using IQ-TREE[29] with model selection7 and 1000 bootstrapping replicates. Consensus phylogenetic trees with branch support values were displayed on Geneious Prime.

### ELISA.
YEZV-positive antigens were prepared from HEK293T cells transfected with a plasmid expressing YEZV N protein under the control of the CAG promoter. HEK293T cells transfected with an empty plasmid were used as a negative antigen. Transfected cells were lysed using the EzRIPA lysis kit (ATTO). Antigens diluted 1:160 in PBS were immobilized on a 96-well Nunc MaxiSorp immuno plate (Thermo Fisher Scientific) followed by blocking using Blocking One (Nacalai Tesque). Serum samples diluted serially or 1:100 were applied for 1 h at room temperature or overnight at 4 °C. Plates were washed three times with PBS containing 0.1% Tween 20 (Merck), then rabbit anti-human IgG H&L (HRP), goat anti-human IgM mu chain (HRP) preadsorbed (Abcam), or recombinant Protein A/G, peroxidase conjugated (Thermo Fisher Scientific) was applied for 1 h at room temperature. Plates were washed three times as above and incubated with the 3,3′,5,5′-tetramethylbenzidine (TMB) liquid substrate system for ELISA peroxidase substrate (Merck). The TMB reaction was terminated by adding 1 M HCl after 15 min incubation and the absorbance at 450 nm of each well was read using an iMark microplate reader (Bio-Rad). A sample was determined as positive when the difference in absorbance between YEZV antigen and mock antigen was >0.3. The reciprocal of the highest serum dilution in which the serum sample was positive was used as the serum antibody titer.

### Ethics statement.
The two patients in the case reports were recruited formally under the research protocols approved by the ethics committee of Sapporo City General Hospital (approval numbers R01-059-615 and R02-059-732) for providing patients' samples to Hokkaido University and the ethical review board of International Institute for Zoonosis Control, Hokkaido University (czc1-4) for using the samples in the present study. The two patients provided written informed consent to publish indirect identifiers. For the retrospective study, samples of 248 patients, who were suspected to be infected with a tick-borne pathogen, were used at Hokkaido Institute of Public Health under the research protocol approved by the human research ethics committee of Hokkaido Institute of Public Health (E19-5, E20-5, and E20-14). Informed consent from these patients was waived based on the Ethical Guidelines for Medical and Health Research Involving Human Subjects by the Ministry of Health, Labour and Welfare, Japan, because (1) the present study uses remnant samples of diagnostic tests, which could be obtained for the study without any invasive procedures and interventions on patients, (2) the remnant samples are anonymized for the present study, meaning that patients enrolled in the present study will not be exposed to any risk, and (3) we announced the present study on the website of Hokkaido Institute of Public Health to allow patients the opportunity to object, as contact between the institute and individual patients is prohibited. The waiver of informed consent has been approved by the ethics committee of Hokkaido Institute of Public Health. The animal experiments were carried out in strict accordance with the Guidelines for Proper Conduct of Animal Experiments of the Science Council of Japan. The protocol was approved by the Hokkaido University Animal Care and Use Committee under approval number 18-0149.

**Reporting summary**. Further information on research design is available in the Nature Research Reporting Summary linked to this article.

## Data availability

All sequences generated in the present study were deposited in the DNA Data Bank of Japan (DDBJ) under the following accession numbers; LC621352–LC621360 and LC628643–LC628645. Source data are provided with this paper.

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

## Acknowledgements

The authors thank Aiko Ohnuma and Koshiro Tabata at the International Institute for Zoonosis Control, Hokkaido University, and Kouichi Tsuzuki, Mayumi Endo, and Kazue Oka in the Laboratory of Microbiology, Faculty of Veterinary Medicine, Hokkaido University for their technical assistance and advise. We appreciate the advice of Noriyo Nagata, Michiyo Kataoka, and Tadaki Suzuki at the National Institute of Infectious Diseases (NIID), Japan, regarding electron microscopy. We are also grateful to Dr. Hiroki Kawabata at the NIID for providing the antigen for the diagnosis of *Borrelia* spp. infections and to Prof. Ulrike Munderloh, University of Minnesota, for permission to use the ISE6 cell line. We thank Edanz (https://jp.edanz.com/ac) for editing a draft of this manuscript.

## Author contributions

F.K., H.Y., K. Matsuno contributed to the conception and design of the study. F.K., H.Y., E.P., K.T., Y.T., K. Mizuma, K. Maeda, K. Matsuno contributed to the data curation and formal analysis. H.Y., L.B.-S., H.S., M. Saijo, K. Matsuno contributed to funding acquisition. H.Y., E.P., K.T., Y.T., K. Mizuma, Y.O., A.G., R.K., M.M., T.I., K.Y., C.F., M.I., A.S., Y.I., L.B.-S., K. Maeda, and K. Matsuno contributed to the investigation and methodology. K. Matsuno administered the project. F.K., M. Sashika, R.N., H.K., K.H., K.O., K. Yoshii, L.B.-S., S.E., A.N., Y.S., H.S., K. Maeda, K. Matsuno contributed to the resources. K. Matsuno contributed to the validation, visualization, and writing the original draft. L.B.-S. and all other authors reviewed and edited the original draft. This work was supported in part by the World-leading Innovative and Smart Education (WISE) Program 1801 from the Ministry of Education, Culture, Sports, Science and Technology (MEXT); the Japan Society for the Promotion of Science/MEXT, grant numbers JP16H06429, JP16H06431, JP16K21723, JP17KT0045, JP19H03112, JP20K18917, JP17H03910; Japan Agency for Medical Research and Development, grant numbers JP19fm0108008, JP21wm0125008, JP21fk0108081; Japan Science and Technology Agency Moonshot R&D, grant number JPMJMS2025; and Akiyama life science foundation. L.B.-S. is supported by the United Kingdom Biotechnology and Biological Sciences Research Council, grant BB/P024270/1. The funders of the study had no role in study design, data collection, data analysis, data interpretation, or writing of the report.

## Competing interests

The authors declare no competing interests.
