## [Peer Review File · Nature Communications]

Reviewer comments, first round -

Reviewers' Comments:

Reviewer #1:

Remarks to the Author:

he manuscript reports a newly discovered tick-borne nairovirus, which appears different from any other known species, and might cause human infections in northern Japan. This study demonstrates that the viruses in the family Nairoviridae continue to emerge as etiologic agents of human disease, and should be included in the differential diagnosis of tick-bitten patients. This research has involved different methods for identifying the human infections, which should be clarified.

1. The gold criterion for validation the human infections of an emerging pathogen is seroconversion or a four-times increase in titres of specific antibodies against the pathogen in the double sera collected between acute and convalescent phase of the illness. Seroconversion has been detected in the two patients. Are there some control population used in the study.
2. In figure 3, Phylogenetic relationship of the M and L genes of orthonairoviruses should be included.
3. In animal infection, the author should provide more detailed information, such as infection route, clinical manifestations, antibody responses, pathological changes, et al., which may further support the virus is associated with human febrile illness.
4. These patients had febrile illness characterized by thrombocytopenia and leukopenia. The authors should discuss another emerging tickborne virus severe fever with thrombocytopenia syndrome virus, which may exist in the studies regions.
5. The format of references should be unified

Reviewer #2:

Remarks to the Author:

The authors of this manuscript have done a thorough investigation of a novel tick-borne virus in Hokkaido Japan. The clinical investigation provides information on two acute cases and lead to the viruses discovery and isolation. Retrospective analysis of archived human samples further revealed previous infections with this virus, designated Yezo virus. Additionally, they have provided evidence of its association with specific tick populations and potential wildlife reservoirs. This is a very complete and detailed manuscript which is well written and highly relevant for publication in Nature Communications. I have only minor points for the authors consideration and that is to include a summary table of the two clinical cases described associating clinical disease manifestations (fever), PCR positivity, serum biochemistries and hematological findings. All the data is present in the manuscript but having it in the context of a larger summary table would allow the reader to see it all at once.

David Safronetz

Reviewer #3:

Remarks to the Author:

A novel nairovirus 1 associated with acute febrile illness in Hokkaido, Japan

The manuscript describes the discovery, isolation and characterization of a novel orthonairovirus, called Yezo virus (YEZV), from two patients with an acute febrile illness in Japan. Retrospectively, a total of 7 patients found to be infected with YEZV from 2014 to 2020 with four of them co-infected with *Borrelia* spp. The consecutive passage of a patient sample in mice and Vero E6 cells resulted in successful virus isolation, which enabled the authors to further characterize the new virus isolate (including via genome sequencing and phylogenetic analyses, growth kinetics and transmission electron microscopy). Interestingly, despite its isolation from patient samples, the

three genome segments phylogenetically clustered with Sulina virus, a tick-derived orthonairovirus from Romania, which is believed to be solely tick-associated. Moreover, serological testing of wildlife and molecular screening of ticks suggest the endemic circulation of YEZV in Japan. This study nicely emphasizes the increasing burden of tick-borne orthonairovirus infections and their overall importance as pathogens with human-pathogenic potential. The manuscript is well-written and the work presented is of significant interest and importance. The big advantage of this study in contrast to many other recent studies on the molecular detection of novel (orthonairo-)viruses is the successful virus isolation and therefore the possibility to further characterize this new virus isolate. Conclusions and claims are reflected by the data and expressed with the necessary care. The description of the methodology is sound and there is enough detail provided for the work to be reproduced.

However, there are a few points that should be addressed:

1. Lines 241 - 242: at which day were the serum samples collected that were used for initial (and unsuccessful) virus isolation attempts? This information may be helpful for any future isolation attempts.
2. Fig. 2 C (growth kinetics): RNA copies per μl are given. However, is there any way to determine virus titres, e.g. through immunofluorescence-based focus assay? Alternatively, is there anything known about the infectivity of this virus? Ratio of infectious to non-infectious virus particles and packaging of genomes? Does the increase in viral RNA simultaneously lead to an increase in infectious virus progeny? Another question would be how the multiplicity of infection was determined? Based on RNA copy numbers?
3. *Haemaphysalis megaspinosa* ticks showed a YEZV detection rate of 3.8%. Serological screening of wildlife revealed a small number of reactive shika deer and raccoons. Considering these findings, it is correct to hypothesize that these species may serve as potential reservoirs. However, please also discuss the limitations of this study given the (partially) reported high serological cross-reactivity among orthonairoviruses and their nucleoproteins, the anticipated circulation of novel, so far unknown orthonairoviruses and the fact that no PCR screening has been performed in these wildlife species.

Taken together, I fully support the publication of this (amended) manuscript because of the originality and importance of the data and the adequate way how these are presented.

Point- by-point response to the reviewers' comments

We really appreciate all three reviewers for their suggestions and comments and tried to answer their suggestions and concerns in the revised manuscript. Line numbers, Tables, and Figures mentioned below are for the revised manuscript file;

Reviewer #1:

The manuscript reports a newly discovered tick-borne nairovirus, which appears different from any other known species, and might cause human infections in northern Japan. This study demonstrates that the viruses in the family Nairoviridae continue to emerge as etiologic agents of human disease, and should be included in the differential diagnosis of tick-bitten patients. This research has involved different methods for identifying the human infections, which should be clarified.

1. The gold criterion for validation the human infections of an emerging pathogen is seroconversion or a four-times increase in titres of specific antibodies against the pathogen in the double sera collected between acute and convalescent phase of the illness. Seroconversion has been detected in the two patients. Are there some control population used in the study.

We agree with the criterion of infections and would like to emphasize the four-times seroconversions of ELISA IgG titers were found in the two patients in the retrospective study (HH007-2016 and HH011-2020 shown in the Table 2) in addition to the two patients in the case reports (HH001-2019 and HH003-2020). To clarify this point, we modified the Table 1 to show IgG ELISA titers of patients at the convalescent phase as well as the text (l. 130-131). During this revision, the number of seroconverted patients was corrected (l. 129). We didn't have control population for the present study because of difficulty to define such population who have not been exposed to the virus.

2. In figure 3, Phylogenetic relationship of the M and L genes of orthonairoviruses should be included.

We moved the phylogenetic trees based on the M and L segments from the Supplementary Figure to the Figure 3 with minor modifications. The text (l. 158-160) and figure legend (l. 503-507) were updated to include these trees.

3. In animal infection, the author should provide more detailed information, such as infection route, clinical manifestations, antibody responses, pathological changes, et al., which may further support

the virus is associated with human febrile illness.

As the reviewer suggested, description of the animal infection experiment was added to the text (l. 137-142). YEZV infection to the immunocompromised mice (AG129 mice) inoculated with the plasma sample of a patient was also confirmed with seroconversion on 14 days after the inoculation, in addition to the virus isolation from serum collected on 5 days after the inoculation. All mice inoculated with the patient plasma did not show any clinical signs in the present study. Since the pathogenicity of YEZV could be affected by potential contaminants in the human sample, we will investigate the pathogenicity of YEZV to mice using the isolated strain in the follow-up study.

4. These patients had febrile illness characterized by thrombocytopenia and leukopenia. The authors should discuss another emerging tickborne virus severe fever with thrombocytopenia syndrome virus, which may exist in the studies regions.

We agree that comparing the characters of YEZV and SFSTV is significantly important since our study could not exclude the possibility of YEZV distribution in the area where SFTSV is endemic. However, because of the limited number of patients in the present study, we could not perform a systematic review for establishing the differentiate diagnosis of these two emerging tick-borne virus infections. We updated the discussion to compare the clinical symptoms and distributions of these two virus diseases (l. 201-207).

5. The format of references should be unified.

Some references were added, and the format was unified.

Reviewer #2:

The authors of this manuscript have done a thorough investigation of a novel tick-borne virus in Hokkaido Japan. The clinical investigation provides information on two acute cases and lead to the viruses discovery and isolation. Retrospective analysis of archived human samples further revealed previous infections with tis virus, designated Yezo virus. Additionally, they have provided evidence of its association with specific tick populations and potential wildlife reservoirs. This is a very complete and detailed manuscript which is well written of highly relevant for publication in Nature Communications. I have only minor points for the authors consideration and that is to include a summary table of the two clinical cases described associating clinical disease manifestations (fever), PCR positivity, serum biochemistries and hematological findings. All the data is present in the

manuscript but having it in the context of a larger summary table would allow the reader to see it all at once.

We appreciate positive comments of this reviewer. We added Table 1 to overview the clinical and laboratory findings on these two patients and renumber the other tables.

Reviewer #3:

A novel nairovirus associated with acute febrile illness in Hokkaido, Japan

The manuscript describes the discovery, isolation and characterization of a novel orthonairovirus, called Yezo virus (YEZV), from two patients with an acute febrile illness in Japan. Retrospectively, a total of 7 patients found to be infected with YEZV from 2014 to 2020 with four of them co-infected with Borrelia spp. The consecutive passage of a patient sample in mice and Vero E6 cells resulted in successful virus isolation, which enabled the authors to further characterize the new virus isolate (including via genome sequencing and phylogenetic analyses, growth kinetics and transmission electron microscopy). Interestingly, despite its isolation from patient samples, the three genome segments phylogenetically clustered with Sulina virus, a tick-derived orthonairovirus from Romania, which is believed to be solely tick-associated. Moreover, serological testing of wildlife and molecular screening of ticks suggest the endemic circulation of YEZV in Japan.

This study nicely emphasizes the increasing burden of tick-borne orthonairovirus infections and their overall importance as pathogens with human-pathogenic potential. The manuscript is well-written and the work presented is of significant interest and importance. The big advantage of this study in contrast to many other recent studies on the molecular detection of novel (orthonairo-)viruses is the successful virus isolation and therefore the possibility to further characterize this new virus isolate. Conclusions and claims are reflected by the data and expressed with the necessary care. The description of the methodology is sound and there is enough detail provided for the work to be reproduced.

However, there are a few points that should be addressed:

1. Lines 241 - 242: at which day were the serum samples collected that were used for initial (and unsuccessful) virus isolation attempts? This information may be helpful for any future isolation attempts.

The serum and urine samples collected on day 5 were used for the trials, and the date was added to the sentence (l. 259). Successful isolation was done from the plasma sample on day 4, collected one day earlier than serum and urine samples. The one-day difference might affect the recovery of infectious virus from the samples (but not mentioned in the text to avoid

misleading).

2. Fig. 2 C (growth kinetics): RNA copies per μ l are given. However, is there any way to determine virus titres, e.g. through immunofluorescence-based focus assay? Alternatively, is there anything known about the infectivity of this virus? Ratio of infectious to non-infectious virus particles and packaging of genomes? Does the increase in viral RNA simultaneously lead to an increase in infectious virus progeny? Another question would be how the multiplicity of infection was determined? Based on RNA copy numbers?

We appreciate the reviewer #3 for raising these truly suggestive questions. We have established a focus-forming assay using patient's convalescent serum to detect virus antigens in infected cells, which allowed us to measure the infectious unit in the virus stock used for inoculation of cells in Fig. 2C (and therefore, m.o.i. was determined). We are sincerely sorry for the lack of description and added this m.o.i. calculation in the Methods section (l. 278-295). Since our claim from the growth kinetics was the susceptibility of each cell line to YEZV replication, RNA copy numbers in the supernatant was determined in the present study, instead of production of infectious virus particles.

As the reviewer #3 questioned, nairovirus-infected cells may produce non-infectious virus particles (such as defective interfering (DI) particles), especially after the inoculation of higher amount of a virus to cells, while virus RNAs lacking a certain region(s) are detected in the supernatants. We thought our growth kinetics study was not the case of producing DI particles, since the m.o.i was lower than 1. However, we believe that it is important to examine the biological characters of YEZV isolates, including production of DI particles, in our future follow-up studies. The discussion section was revisited to emphasize that the present study was very limited to the basic characterization of YEZV (l. 233-236).

3. Haemaphysalis megaspinosa ticks showed a YEZV detection rate of 3.8%. Serological screening of wildlife revealed a small number of reactive shika deer and raccoons. Considering these findings, it is correct to hypothesize that these species may serve as potential reservoirs. However, please also discuss the limitations of this study given the (partially) reported high serological cross-reactivity among orthonairoviruses and their nucleoproteins, the anticipated circulation of novel, so far unknown orthonairoviruses and the fact that no PCR screening has been performed in these wildlife species.

We completely agree with this comment. A sentence explaining the limitation of the study design was added to the discussion (l. 220-222).

Taken together, I fully support the publication of this (amended) manuscript because of the originality and importance of the data and the adequate way how these are presented.

We sincerely thank positive comments.

Reviewer comments, second round -

Reviewer #1 (Remarks to the Author):

The authors have substantially revised the paper, but the format of references should be unified, and the full manuscript can be improved by a native English speaker.

There some minor errors that should be modified.

1.Line 8: have been found.

2. Line 12-13. YEZV was phylogenetically grouped with an orthonairovirus, Sulina virus, detected in Ixodes ricinus ticks in Romania, which was initially considered to be a tick-specific virus.

3. Line 14, during 2014-2020.

4. Line 29, no associated disease.

5.Line 32, described.

Reviewer #2 (Remarks to the Author):

The authors have addressed my comments.

Reviewer #3 (Remarks to the Author):

The authors have satisfactorily addressed all questions and comments.

Point-by-point response to the reviewers' comments

Again, we are grateful to all three reviewers for their suggestions and comments to improve our manuscript.

Reviewer #1:

The authors have substantially revised the paper, but the format of references should be unified, and the full manuscript can be improved by a native English speaker.

We are sorry for the inconsistency of presentation in our References section. We believe in this version, all formatting issues have been addressed in the revised manuscript. The manuscript has also been edited by a native English speaker coauthor and an English editing service.

There are some minor errors that should be modified.

1.Line 8: have been found.

2. Line 12-13. YEZV was phylogenetically grouped with an orthonairovirus, Sulina virus, detected in Ixodes ricinus ticks in Romania, which was initially considered to be a tick-specific virus.

3. Line 14, during 2014-2020.

4. Line 29, no associated disease.

5.Line 32, described.

We appreciate the correction of these minor errors, and these changes have now been applied to the revised manuscript. Error number #2 has been resolved in formatting process to adjust the abstract to the word limit.